# Recurrent Batch Normalization

**Tim Cooijmans, Nicolas Ballas, César Laurent, Çağlar Gülçehre & Aaron Courville**
MILA - Université de Montréal
`firstname.lastname@umontreal.ca`

## Abstract

We propose a reparameterization of LSTM that brings the benefits of batch normalization to recurrent neural networks. Whereas previous works only apply batch normalization to the input-to-hidden transformation of RNNs, we demonstrate that it is both possible and beneficial to batch-normalize the hidden-to-hidden transition, thereby reducing internal covariate shift between time steps.

We evaluate our proposal on various sequential problems such as sequence classification, language modeling and question answering. Our empirical results show that our batch-normalized LSTM consistently leads to faster convergence and improved generalization.

## 1 Introduction

Recurrent neural network architectures such as LSTM (Hochreiter & Schmidhuber, 1997) and GRU (Cho et al., 2014) have recently exhibited state-of-the-art performance on a wide range of complex sequential problems including speech recognition Amodei et al. (2015), machine translation (Bahdanau et al., 2015) and image and video captioning (Xu et al., 2015; Yao et al., 2015). Top-performing models, however, are based on very high-capacity networks that are computationally intensive and costly to train. Effective optimization of recurrent neural networks is thus an active area of study (Pascanu et al., 2012; Martens & Sutskever, 2011; Ollivier, 2013).

It is well-known that for deep feed-forward neural networks, covariate shift (Shimodaira, 2000; Ioffe & Szegedy, 2015) degrades the efficiency of training. Covariate shift is a change in the distribution of the inputs to a model. This occurs continuously during training of feed-forward neural networks, where changing the parameters of a layer affects the distribution of the inputs to all layers above it. As a result, the upper layers are continually adapting to the shifting input distribution and unable to learn effectively. This *internal* covariate shift (Ioffe & Szegedy, 2015) may play an especially important role in recurrent neural networks, which resemble very deep feed-forward networks.

Batch normalization (Ioffe & Szegedy, 2015) is a recently proposed technique for controlling the distributions of feed-forward neural network activations, thereby reducing internal covariate shift. It involves standardizing the activations going into each layer, enforcing their means and variances to be invariant to changes in the parameters of the underlying layers. This effectively decouples each layer's parameters from those of other layers, leading to a better-conditioned optimization problem. Indeed, deep neural networks trained with batch normalization converge significantly faster and generalize better.

Although batch normalization has demonstrated significant training speed-ups and generalization benefits in feed-forward networks, it is proven to be difficult to apply in recurrent architectures (Laurent et al., 2016; Amodei et al., 2015). It has found limited use in stacked RNNs, where the normalization is applied "vertically", i.e. to the input of each RNN, but not "horizontally" between timesteps. RNNs are deeper in the time direction, and as such batch normalization would be most beneficial when applied horizontally. However, Laurent et al. (2016) hypothesized that applying batch normalization in this way hurts training because of exploding gradients due to repeated rescaling.

Our findings run counter to this hypothesis. We show that it is both possible and highly beneficial to apply batch normalization in the hidden-to-hidden transition of recurrent models. In particular, we describe a reparameterization of LSTM (Section 3) that involves batch normalization and demonstrate that it is easier to optimize and generalizes better. In addition, we empirically analyze the

gradient backpropagation and show that proper initialization of the batch normalization parameters is crucial to avoiding vanishing gradient (Section 4). We evaluate our proposal on several sequential problems and show (Section 5) that our LSTM reparameterization consistently outperforms the LSTM baseline across tasks, in terms of both time to convergence and performance.

Liao & Poggio (2016) simultaneously investigated batch normalization in recurrent neural networks, albeit only for very short sequences (10 steps). Ba et al. (2016) independently developed a variant of batch normalization that is also applicable to recurrent neural networks and delivers similar improvements as our method.

## 2 PREREQUISITES

### 2.1 LSTM

Long Short-Term Memory (LSTM) networks are an instance of a more general class of recurrent neural networks (RNNs), which we review briefly in this paper. Given an input sequence $\mathbf{X} = (\mathbf{x}_1, \mathbf{x}_2, \ldots, \mathbf{x}_T)$, an RNN defines a sequence of hidden states $\mathbf{h}_t$ according to

$$\mathbf{h}_t = \phi(\mathbf{W}_h \mathbf{h}_{t-1} + \mathbf{W}_x \mathbf{x}_t + \mathbf{b}), \tag{1}$$

where $\mathbf{W}_h \in \mathbb{R}^{d_h \times d_h}, \mathbf{W}_x \in \mathbb{R}^{d_x \times d_h}, \mathbf{b} \in \mathbb{R}^{d_h}$ and the initial state $\mathbf{h}_0 \in \mathbb{R}^{d_h}$ are model parameters. A popular choice for the activation function $\phi(\,\cdot\,)$ is tanh.

RNNs are popular in sequence modeling thanks to their natural ability to process variable-length sequences. However, training RNNs using first-order stochastic gradient descent (SGD) is notoriously difficult due to the well-known problem of exploding/vanishing gradients (Bengio et al., 1994; Hochreiter, 1991; Pascanu et al., 2012). Gradient vanishing occurs when states $\mathbf{h}_t$ are not influenced by small changes in much earlier states $\mathbf{h}_\tau, t \ll \tau$, preventing learning of long-term dependencies in the input data. Although learning long-term dependencies is fundamentally difficult (Bengio et al., 1994), its effects can be mitigated through architectural variations such as LSTM (Hochreiter & Schmidhuber, 1997), GRU (Cho et al., 2014) and $i$RNN/$u$RNN (Le et al., 2015; Arjovsky et al., 2015).

In what follows, we focus on the LSTM architecture (Hochreiter & Schmidhuber, 1997) with recurrent transition given by

$$\begin{pmatrix} \tilde{\mathbf{f}}_t \\ \tilde{\mathbf{i}}_t \\ \tilde{\mathbf{o}}_t \\ \tilde{\mathbf{g}}_t \end{pmatrix} = \mathbf{W}_h \mathbf{h}_{t-1} + \mathbf{W}_x \mathbf{x}_t + \mathbf{b} \tag{2}$$

$$\mathbf{c}_t = \sigma(\tilde{\mathbf{f}}_t) \odot \mathbf{c}_{t-1} + \sigma(\tilde{\mathbf{i}}_t) \odot \tanh(\tilde{\mathbf{g}}_t) \tag{3}$$

$$\mathbf{h}_t = \sigma(\tilde{\mathbf{o}}_t) \odot \tanh(\mathbf{c}_t), \tag{4}$$

where $\mathbf{W}_h \in \mathbb{R}^{d_h \times 4d_h}, \mathbf{W}_x \in \mathbb{R}^{d_x \times 4d_h}, \mathbf{b} \in \mathbb{R}^{4d_h}$ and the initial states $\mathbf{h}_0 \in \mathbb{R}^{d_h}, \mathbf{c}_0 \in \mathbb{R}^{d_h}$ are model parameters. $\sigma$ is the logistic sigmoid function, and the $\odot$ operator denotes the Hadamard product.

The LSTM differs from simple RNNs in that it has an additional memory *cell* $\mathbf{c}_t$ whose update is nearly linear which allows the gradient to flow back through time more easily. In addition, unlike the RNN which overwrites its content at each timestep, the update of the LSTM cell is regulated by a set of gates. The forget gate $\mathbf{f}_t$ determines the extent to which information is carried over from the previous timestep, and the input gate $\mathbf{i}_t$ controls the flow of information from the current input $\mathbf{x}_t$. The output gate $\mathbf{o}_t$ allows the model to read from the cell. This carefully controlled interaction with the cell is what allows the LSTM to robustly retain information for long periods of time.

### 2.2 BATCH NORMALIZATION

*Covariate shift* (Shimodaira, 2000) is a phenomenon in machine learning where the features presented to a model change in distribution. In order for learning to succeed in the presence of covariate shift, the model's parameters must be adjusted not just to learn the concept at hand but also to adapt to the changing distribution of the inputs. In deep neural networks, this problem manifests as

*internal covariate shift* (Ioffe & Szegedy, 2015), where changing the parameters of a layer affects the distribution of the inputs to all layers above it.

Batch Normalization (Ioffe & Szegedy, 2015) is a recently proposed network reparameterization which aims to reduce internal covariate shift. It does so by standardizing the activations using empirical estimates of their means and standard deviations. However, it does not decorrelate the activations due to the computationally costly matrix inversion. The batch normalizing transform is as follows:

$$\text{BN}(\mathbf{h}; \gamma, \beta) = \beta + \gamma \odot \frac{\mathbf{h} - \widehat{\mathbb{E}}[\mathbf{h}]}{\sqrt{\widehat{\text{Var}}[\mathbf{h}] + \epsilon}} \tag{5}$$

where $\mathbf{h} \in \mathbb{R}^d$ is the vector of (pre)activations to be normalized, $\gamma \in \mathbb{R}^d, \beta \in \mathbb{R}^d$ are model parameters that determine the mean and standard deviation of the normalized activation, and $\epsilon \in \mathbb{R}$ is a regularization hyperparameter. The division should be understood to proceed elementwise.

At training time, the statistics $\mathbb{E}[\mathbf{h}]$ and $\text{Var}[\mathbf{h}]$ are estimated by the sample mean and sample variance of the current minibatch. This allows for backpropagation through the statistics, preserving the convergence properties of stochastic gradient descent. During inference, the statistics are typically estimated based on the entire training set, so as to produce a deterministic prediction.

## 3 BATCH-NORMALIZED LSTM

This section introduces a reparameterization of LSTM that takes advantage of batch normalization. Contrary to Laurent et al. (2016); Amodei et al. (2015), we leverage batch normalization in both the input-to-hidden *and* the hidden-to-hidden transformations. We introduce the batch-normalizing transform $\text{BN}(\cdot; \gamma, \beta)$ into the LSTM as follows:

$$\begin{pmatrix} \tilde{\mathbf{f}}_t \\ \tilde{\mathbf{i}}_t \\ \tilde{\mathbf{o}}_t \\ \tilde{\mathbf{g}}_t \end{pmatrix} = \text{BN}(\mathbf{W}_h \mathbf{h}_{t-1}; \gamma_h, \beta_h) + \text{BN}(\mathbf{W}_x \mathbf{x}_t; \gamma_x, \beta_x) + \mathbf{b} \tag{6}$$

$$\mathbf{c}_t = \sigma(\tilde{\mathbf{f}}_t) \odot \mathbf{c}_{t-1} + \sigma(\tilde{\mathbf{i}}_t) \odot \tanh(\tilde{\mathbf{g}}_t) \tag{7}$$

$$\mathbf{h}_t = \sigma(\tilde{\mathbf{o}}_t) \odot \tanh(\text{BN}(\mathbf{c}_t; \gamma_c, \beta_c)) \tag{8}$$

In our formulation, we normalize the recurrent term $\mathbf{W}_h \mathbf{h}_{t-1}$ and the input term $\mathbf{W}_x \mathbf{x}_t$ separately. Normalizing these terms individually gives the model better control over the relative contribution of the terms using the $\gamma_h$ and $\gamma_x$ parameters. We set $\beta_h = \beta_x = \mathbf{0}$ to avoid unnecessary redundancy, instead relying on the pre-existing parameter vector $\mathbf{b}$ to account for both biases. In order to leave the LSTM dynamics intact and preserve the gradient flow through $\mathbf{c}_t$, we do not apply batch normalization in the cell update.

The batch normalization transform relies on batch statistics to standardize the LSTM activations. It would seem natural to share the statistics that are used for normalization across time, just as recurrent neural networks share their parameters over time. However, we find that simply averaging statistics over time severely degrades performance. Although LSTM activations do converge to a stationary distribution, we observe that their statistics during the initial transient differ significantly (see Figure 5 in Appendix A). Consequently, we recommend using separate statistics for each timestep to preserve information of the initial transient phase in the activations.[1]

Generalizing the model to sequences longer than those seen during training is straightforward thanks to the rapid convergence of the activations to their steady-state distributions (cf. Figure 5). For our experiments we estimate the population statistics separately for each timestep $1, \ldots, T_{max}$ where

---

[1] Note that we separate *only* the statistics over time and not the $\gamma$ and $\beta$ parameters.

$T_{max}$ is the length of the longest training sequence. When at test time we need to generalize beyond $T_{max}$, we use the population statistic of time $T_{max}$ for all time steps beyond it.

During training we estimate the statistics across the minibatch, independently for each timestep. At test time we use estimates obtained by averaging the minibatch estimates over the training set.

## 4 INITIALIZING $\gamma$ FOR GRADIENT FLOW

Although batch normalization allows for easy control of the pre-activation variance through the $\gamma$ parameters, common practice is to normalize to unit variance. We suspect that the previous difficulties with recurrent batch normalization reported in Laurent et al. (2016); Amodei et al. (2015) are largely due to improper initialization of the batch normalization parameters, and $\gamma$ in particular. In this section we demonstrate the impact of $\gamma$ on gradient flow.

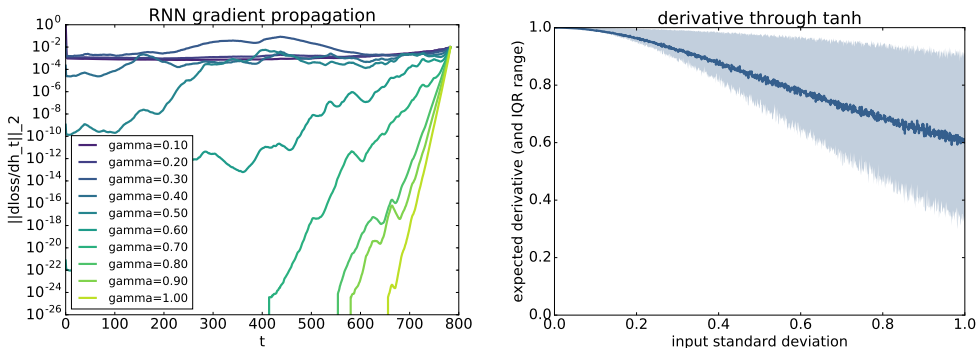

(a) We visualize the gradient flow through a batch-normalized $\tanh$ RNN as a function of $\gamma$. High variance causes vanishing gradient.

(b) We show the empirical expected derivative and interquartile range of $\tanh$ nonlinearity as a function of input variance. High variance causes saturation, which decreases the expected derivative.

Figure 1: Influence of pre-activation variance on gradient propagation.

In Figure 1(a), we show how the pre-activation variance impacts gradient propagation in a simple RNN on the sequential MNIST task described in Section 5.1. Since backpropagation operates in reverse, the plot is best read from right to left. The quantity plotted is the norm of the gradient of the loss with respect to the hidden state at different time steps. For large values of $\gamma$, the norm quickly goes to zero as gradient is propagated back in time. For small values of $\gamma$ the norm is nearly constant.

To demonstrate what we think is the cause of this vanishing, we drew samples $x$ from a set of centered Gaussian distributions with standard deviation ranging from 0 to 1, and computed the derivative $\tanh'(x) = 1 - \tanh^2(x) \in [0, 1]$ for each. Figure 1(b) shows the empirical distribution of the derivative as a function of standard deviation. When the input standard deviation is low, the input tends to be close to the origin where the derivative is close to 1. As the standard deviation increases, the expected derivative decreases as the input is more likely to be in the saturation regime. At unit standard deviation, the expected derivative is much smaller than 1.

We conjecture that this is what causes the gradient to vanish, and recommend initializing $\gamma$ to a small value. In our trials we found that values of 0.01 or lower caused instabilities during training. Our choice of 0.1 seems to work well across different tasks.

## 5 EXPERIMENTS

This section presents an empirical evaluation of the proposed batch-normalized LSTM on four different tasks. Note that for all the experiments, we initialize the batch normalization scale and shift parameters $\gamma$ and $\beta$ to 0.1 and 0 respectively.

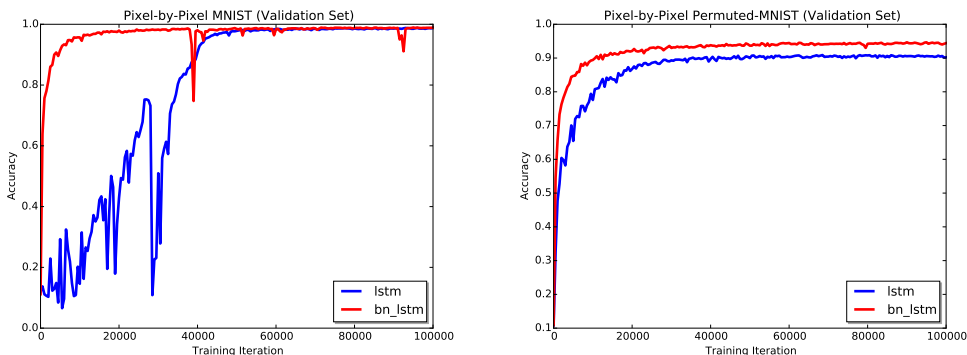

Figure 2: Accuracy on the validation set for the pixel by pixel MNIST classification tasks. The batch-normalized LSTM is able to converge faster relatively to a baseline LSTM. Batch-normalized LSTM also shows some improve generalization on the permuted sequential MNIST that require to preserve long-term memory information.

## 5.1 SEQUENTIAL MNIST

We evaluate our batch-normalized LSTM on a sequential version of the MNIST classification task (Le et al., 2015). The model processes each image one pixel at a time and finally predicts the label. We consider both sequential MNIST tasks, MNIST and permuted MNIST ($p$MNIST). In MNIST, the pixels are processed in scanline order. In $p$MNIST the pixels are processed in a fixed random order.

Our baseline consists of an LSTM with 100 hidden units, with a softmax classifier to produce a prediction from the final hidden state. We use orthogonal initialization for all weight matrices, except for the hidden-to-hidden weight matrix which we initialize to be the identity matrix, as this yields better generalization performance on this task for both models. The model is trained using RMSProp (Tieleman & Hinton, 2012) with learning rate of $10^{-3}$ and $0.9$ momentum. We apply gradient clipping at 1 to avoid exploding gradients.

The in-order MNIST task poses a unique problem for our model: the input for the first hundred or so timesteps is constant across examples since the upper pixels are almost always black. This causes the variance of the hidden states to be exactly zero for a long period of time. Normalizing these zero-variance activations involves dividing zero by a small number at many timesteps, which does not affect the forward-propagated activations but causes the back-propagated gradient to explode. We work around this by adding Gaussian noise to the initial hidden states. Although the normalization amplifies the noise to signal level, we find that it does not hurt performance compared to data-dependent ways of initializing the hidden states.

| Model | MNIST | $p$MNIST |
|---|---|---|
| TANH-RNN (Le et al., 2015) | 35.0 | 35.0 |
| $i$RNN (Le et al., 2015) | 97.0 | 82.0 |
| $u$RNN (Arjovsky et al., 2015) | 95.1 | 91.4 |
| $s$TANH-RNN (Zhang et al., 2016) | 98.1 | 94.0 |
| LSTM (ours) | 98.9 | 90.2 |
| BN-LSTM (ours) | **99.0** | **95.4** |

Table 1: Accuracy obtained on the test set for the pixel by pixel MNIST classification tasks

In Figure 2 we show the validation accuracy while training for both LSTM and batch-normalized LSTM (BN-LSTM). BN-LSTM converges faster than LSTM on both tasks. Additionally, we observe that BN-LSTM generalizes significantly better on $p$MNIST. It has been highlighted in Arjovsky et al. (2015) that $p$MNIST contains many longer term dependencies across pixels than in the original pixel ordering, where a lot of structure is local. A recurrent network therefore needs to

| Model | Penn Treebank |
|---|---|
| LSTM (Graves, 2013) | 1.26[2] |
| HF-MRNN (Mikolov et al., 2012) | 1.41 |
| Norm-stabilized LSTM (Krueger & Memisevic, 2016) | 1.39 |
| ME n-gram (Mikolov et al., 2012) | 1.37 |
| LSTM (ours) | 1.38 |
| BN-LSTM (ours) | 1.32 |
| Zoneout (Krueger et al., 2016) | 1.27 |
| HM-LSTM (Chung et al., 2016) | 1.24 |
| HyperNetworks (Ha et al., 2016) | **1.22** |

Table 2: Bits-per-character on the Penn Treebank test sequence.

characterize dependencies across varying time scales in order to solve this task. Our results suggest that BN-LSTM is better able to capture these long-term dependencies.

Table 1 reports the test set accuracy of the early stop model for LSTM and BN-LSTM using the population statistics. Recurrent batch normalization leads to a better test score, especially for $p$MNIST where models have to leverage long-term temporal depencies. In addition, Table 1 shows that our batch-normalized LSTM achieves state of the art on both MNIST and $p$MNIST.

## 5.2 CHARACTER-LEVEL PENN TREEBANK

We evaluate our model on the task of character-level language modeling on the Penn Treebank corpus (Marcus et al., 1993) according to the train/valid/test partition of Mikolov et al. (2012). For training, we segment the training sequence into examples of length 100. The training sequence does not cleanly divide by 100, so for each epoch we randomly crop a subsequence that does and segment that instead.

Our baseline is an LSTM with 1000 units, trained to predict the next character using a softmax classifier on the hidden state $h_t$. We use stochastic gradient descent on minibatches of size 64, with gradient clipping at 1.0 and step rule determined by Adam (Kingma & Ba, 2014) with learning rate 0.002. We use orthogonal initialization for all weight matrices. The setup for the batch-normalized LSTM is the same in all respects except for the introduction of batch normalization as detailed in 3.

We show the learning curves in Figure 3(a). BN-LSTM converges faster and generalizes better than the LSTM baseline. Figure 3(b) shows the generalization of our model to longer sequences. We observe that using the population statistics improves generalization performance, which confirms that repeating the last population statistic (cf. Section 3) is a viable strategy. In table 2 we report the performance of our best models (early-stopped on validation performance) on the Penn Treebank test sequence. Follow up works havd since improved the state of the art (Krueger et al., 2016; Chung et al., 2016; Ha et al., 2016).

## 5.3 TEXT8

We evaluate our model on a second character-level language modeling task on the much larger text8 dataset (Mahoney, 2009). This dataset is derived from Wikipedia and consists of a sequence of 100M characters including only alphabetical characters and spaces. We follow Mikolov et al. (2012); Zhang et al. (2016) and use the first 90M characters for training, the next 5M for validation and the final 5M characters for testing. We train on nonoverlapping sequences of length 180.

Both our baseline and batch-normalized models are LSTMs with 2000 units, trained to predict the next character using a softmax classifier on the hidden state $h_t$. We use stochastic gradient descent on minibatches of size 128, with gradient clipping at 1.0 and step rule determined by Adam (Kingma & Ba, 2014) with learning rate 0.001. All weight matrices were initialized to be orthogonal.

We early-stop on validation performance and report the test performance of the resulting model in table 3. We observe that BN-LSTM obtains a significant performance improvement over the LSTM baseline. Chung et al. (2016) has since improved on our performance.

| Model | text8 |
|---|---|
| $td$-LSTM (Zhang et al., 2016) | 1.63 |
| HF-MRNN (Mikolov et al., 2012) | 1.54 |
| skipping RNN (Pachitariu & Sahani, 2013) | 1.48 |
| LSTM (ours) | 1.43 |
| BN-LSTM (ours) | 1.36 |
| HM-LSTM (Chung et al., 2016) | **1.29** |

Table 3: Bits-per-character on the text8 test sequence.

## 5.4 TEACHING MACHINES TO READ AND COMPREHEND

Recently, Hermann et al. (2015) introduced a set of challenging benchmarks for natural language processing, along with neural network architectures to address them. The tasks involve reading real news articles and answering questions about their content. Their principal model, the Attentive Reader, is a recurrent neural network that invokes an attention mechanism to locate relevant information in the document. Such models are notoriously hard to optimize and yet increasingly popular.

To demonstrate the generality and practical applicability of our proposal, we apply batch normalization in the Attentive Reader model and show that this drastically improves training.

We evaluate several variants. The first variant, referred to as BN-LSTM, consists of the vanilla Attentive Reader model with the LSTM simply replaced by our BN-LSTM reparameterization. The second variant, termed BN-everywhere, is exactly like the first, except that we also introduce batch normalization into the attention computations, normalizing each term going into the $\tanh$ nonlinearities.

Our third variant, BN-e*, is like BN-everywhere, but improved to more carefully handle variable-length sequences. Throughout this experiment we followed the common practice of padding each batch of variable-length data with zeros. However, this biases the batch mean and variance of $\mathbf{x}_t$ toward zero. We address this effect using *sequencewise* normalization of the inputs as proposed by Laurent et al. (2016); Amodei et al. (2015). That is, we share statistics over time for normalization

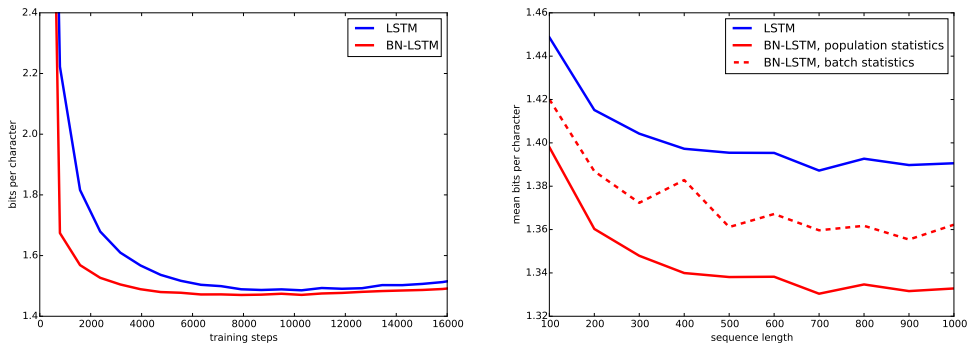

(a) Performance in bits-per-character on length-100 subsequences of the Penn Treebank validation sequence during training.

(b) Generalization to longer subsequences of Penn Treebank using population statistics. The subsequences are taken from the test sequence.

Figure 3: Penn Treebank evaluation

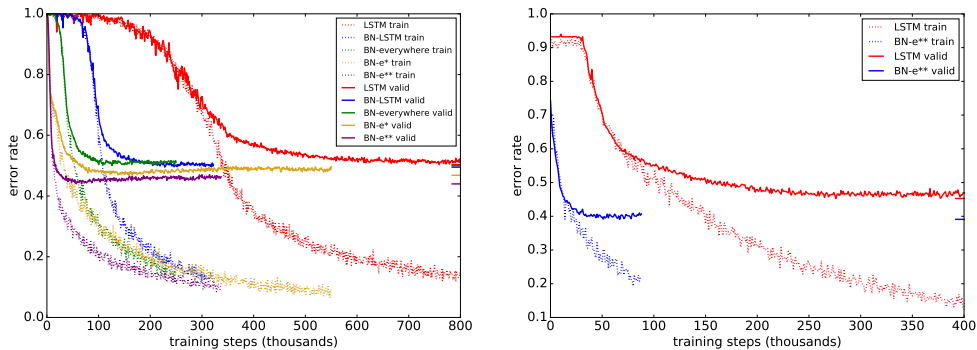

(a) Error rate on the validation set for the Attentive Reader models on a variant of the CNN QA task (Hermann et al., 2015). As detailed in Appendix C, the theoretical lower bound on the error rate on this task is 43%.

(b) Error rate on the validation set on the full CNN QA task from Hermann et al. (2015).

Figure 4: Training curves on the CNN question-answering tasks.

of the input terms $\mathbf{W}_x\mathbf{x}_t$, but *not* for the recurrent terms $\mathbf{W}_h\mathbf{h}_t$ or the cell output $\mathbf{c}_t$. Doing so avoids many issues involving degenerate statistics due to input sequence padding.

Our fourth and final variant BN-e** is like BN-e* but bidirectional. The main difficulty in adapting to bidirectional models also involves padding. Padding poses no problem as long as it is properly ignored (by not updating the hidden states based on padded regions of the input). However to perform the reverse application of a bidirectional model, it is common to simply reverse the padded sequences, thus moving the padding to the front. This causes similar problems as were observed on the sequential MNIST task (Section 5.1): the hidden states will not diverge during the initial timesteps and hence their variance will be severely underestimated. To get around this, we reverse only the unpadded portion of the input sequences and leave the padding in place.

See Appendix C for hyperparameters and task details.

Figure 4(a) shows the learning curves for the different variants of the attentive reader. BN-LSTM trains dramatically faster than the LSTM baseline. BN-everywhere in turn shows a significant improvement over BN-LSTM. In addition, both BN-LSTM and BN-everywhere show a generalization benefit over the baseline. The validation curves have minima of 50.3%, 49.5% and 50.0% for the baseline, BN-LSTM and BN-everywhere respectively. We emphasize that these results were obtained without any tweaking – all we did was to introduce batch normalization.

BN-e* and BN-e** converge faster yet, and reach lower minima: 47.1% and 43.9% respectively.

| Model | CNN valid | CNN test |
|---|---|---|
| Attentive Reader (Hermann et al., 2015) | 38.4 | 37.0 |
| LSTM (ours) | 45.5 | 45.0 |
| BN-e** (ours) | **37.9** | **36.3** |

Table 4: Error rates on the CNN question-answering task Hermann et al. (2015).

We train and evaluate our best model, BN-e**, on the full task from (Hermann et al., 2015). On this dataset we had to reduce the number of hidden units to 120 to avoid severe overfitting. Training curves for BN-e** and a vanilla LSTM are shown in Figure 4(b). Table 4 reports performances of the early-stopped models.

# 6 CONCLUSION

Contrary to previous findings by Laurent et al. (2016); Amodei et al. (2015), we have demonstrated that batch-normalizing the hidden states of recurrent neural networks greatly improves optimization. Indeed, doing so yields benefits similar to those of batch normalization in feed-forward neural networks: our proposed BN-LSTM trains faster and generalizes better on a variety of tasks including language modeling and question-answering. We have argued that proper initialization of the batch normalization parameters is crucial, and suggest that previous difficulties (Laurent et al., 2016; Amodei et al., 2015) were due in large part to improper initialization. Finally, we have shown our model to apply to complex settings involving variable-length data, bidirectionality and highly nonlinear attention mechanisms.

## ACKNOWLEDGEMENTS

The authors would like to acknowledge the following agencies for research funding and computing support: the Nuance Foundation, Samsung, NSERC, Calcul Québec, Compute Canada, the Canada Research Chairs and CIFAR. Experiments were carried out using the Theano (Team et al., 2016) and the Blocks and Fuel (van Merriënboer et al., 2015) libraries for scientific computing. We thank David Krueger, Saizheng Zhang, Ishmael Belghazi and Yoshua Bengio for discussions and suggestions.

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

## A   CONVERGENCE OF POPULATION STATISTICS

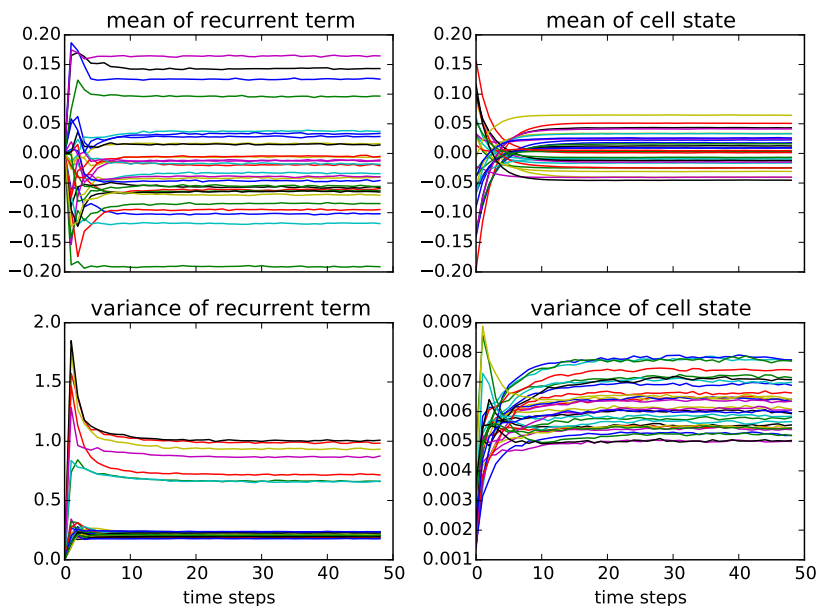

Figure 5: Convergence of population statistics to stationary distributions on the Penn Treebank task. The horizontal axis denotes RNN time. Each curve corresponds to a single hidden unit. Only a random subset of units is shown. See Section 3 for discussion.

## B   SENSITIVITY TO INITIALIZATION OF $\gamma$

In Section 4 we investigated the effect of initial $\gamma$ on gradient flow. To show the practical implications of this, we performed several experiments on the $p$MNIST and Penn Treebank benchmarks. The resulting performances are shown in Figure 6.

The $p$MNIST training curves confirm that higher initial values of $\gamma$ are detrimental to the optimization of the model. For the Penn Treebank task however, the effect is gone.

We believe this is explained by the difference in the nature of the two tasks. For $p$MNIST, the model absorbs the input sequence and only at the end of the sequence does it make a prediction on which it receives feedback. Learning from this feedback requires propagating the gradient all the way back through the sequence.

In the Penn Treebank task on the other hand, the model makes a prediction at each timestep. At each step of the backward pass, a fresh learning signal is added to the backpropagated gradient. Essentially, the model is able to get off the ground by picking up short-term dependencies. This fails on $p$MNIST wich is dominated by long-term dependencies (Arjovsky et al., 2015).

## C   TEACHING MACHINES TO READ AND COMPREHEND: TASK SETUP

We evaluate the models on the question answering task using the CNN corpus (Hermann et al., 2015), with placeholders for the named entities. We follow a similar preprocessing pipeline as Hermann et al. (2015). During training, we randomly sample the examples with replacement and shuffle the order of the placeholders in each text inside the minibatch. We use a vocabulary of 65829 words.

We deviate from Hermann et al. (2015) in order to save computation: we use only the 4 most relevant sentences from the description, as identified by a string matching procedure. Both the training and validation sets are preprocessed in this way. Due to imprecision this heuristic sometimes strips the

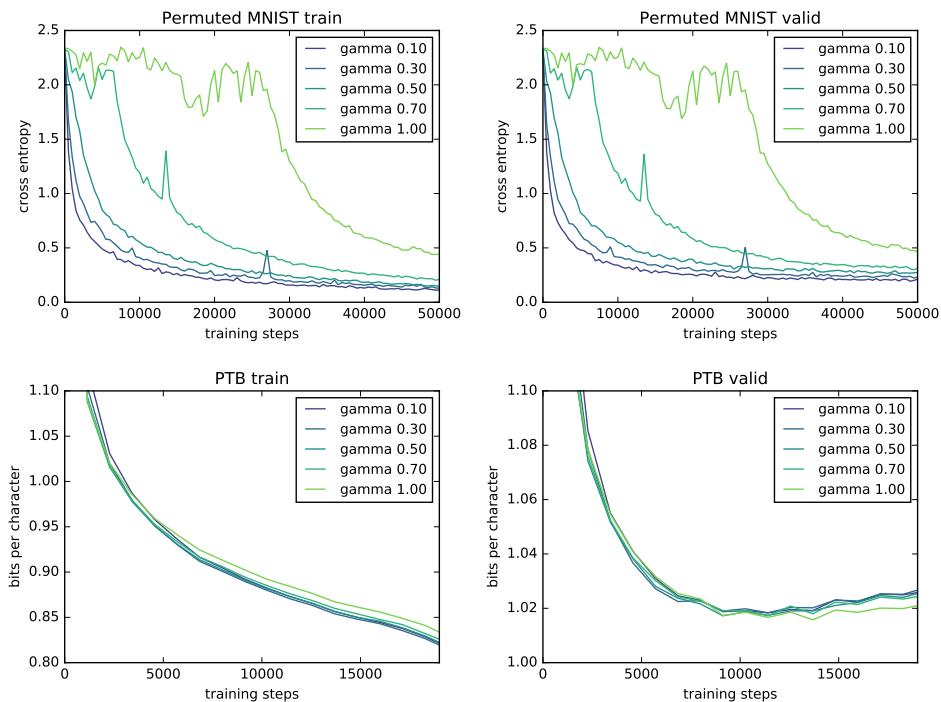

Figure 6: Training curves on $p$MNIST and Penn Treebank for various initializations of $\gamma$.

answers from the passage, putting an upper bound of 57% on the validation accuracy that can be achieved.

For the reported performances, the first three models (LSTM, BN-LSTM and BN-everywhere) are trained using the exact same hyperparameters, which were chosen because they work well for the baseline. The hidden state is composed of 240 units. We use stochastic gradient descent on mini-batches of size 64, with gradient clipping at 10 and step rule determined by Adam (Kingma & Ba, 2014) with learning rate $8 \times 10^{-5}$.

For BN-e* and BN-e**, we use the same hyperparameters except that we reduce the learning rate to $8 \times 10^{-4}$ and the minibatch size to 40.

## D    HYPERPARAMETER SEARCHES

Table 5 reports hyperparameter values that were tried in the experiments.

<table>
<tr><td colspan="2">(a) MNIST and $p$MNIST</td><td colspan="2">(b) Penn Treebank</td></tr>
<tr><td>Learning rate:</td><td>1e-2, 1e-3, 1e-4</td><td>Learning rate:</td><td>1e-1, 1e-2, 2e-2, 1e-3</td></tr>
<tr><td>RMSProp momentum:</td><td>0.5, 0.9</td><td>Hidden state size:</td><td>800, 1000, 1200, 1500, 2000</td></tr>
<tr><td>Hidden state size:</td><td>100, 200, 400</td><td>Batch size:</td><td>32, 64, 100, 128</td></tr>
<tr><td>Initial $\gamma$:</td><td>1e-1, 3e-1, 5e-1, 7e-1, 1.0</td><td>Initial $\gamma$:</td><td>1e-1, 3e-1, 5e-1, 7e-1, 1.0</td></tr>
<tr><td colspan="2">(c) Text8</td><td colspan="2">(d) Attentive Reader</td></tr>
<tr><td>Learning rate:</td><td>1e-1, 1e-2, 1e-3</td><td>Learning rate:</td><td>8e-3, 8e-4, 8e-5, 8e-6</td></tr>
<tr><td>Hidden state size:</td><td>500, 1000, 2000, 4000</td><td>Hidden state size:</td><td>60, 120, 240, 280</td></tr>
</table>

Table 5: Hyperparameter values that have been explored in the experiments.

For MNIST and $p$MNIST, the hyperparameters were varied independently. For Penn Treebank, we performed a full grid search on learning rate and hidden state size, and later performed a sensitivity

analysis on the batch size and initial $\gamma$. For the text8 task and the experiments with the Attentive Reader, we carried out a grid search on the learning rate and hidden state size.

The same values were tried for both the baseline and our BN-LSTM. In each case, our reported results are those of the model with the best validation performance.

