# Peer review of "Recurrent Batch Normalization"

_ICLR 2017 — accepted_

[Official Review · AnonReviewer2 · rating 7 · confidence 4 · 16 Dec 2016]
**Review: Recurrent Batch Normalization**

Contributions
The paper presents an adaptation of batch normalization for RNNs in the case of LSTMs, along the horizontal depth. Contrary to previous work from (Laurent 2015; Amodei 2016), the work demonstrates that batch-normalizing the hidden states of RNNs can improve optimization, and argues with quantitative experiments that the key factor to making this work is proper initialization of parameters, in particular gamma. Experiments show some gain in performance over vanilla LSTMs on Sequential MNIST, PTB, Text8 and CNN Question-Answering.

Novelty+Significance
Batch normalization has been key for training deeper and deeper networks (e.g. ResNets) and it seems natural that we would want to extend it to RNNs.  The paper shows that it is possible to do so with proper initialization of parameters, contrary to previous work from (Laurent 2015; Amodei 2016). Novelty comes from where to batch norm (i.e. not in the cell update) and in the per-time step statistics. 

Adding batch normalization to LSTMs incurs additional computational cost and bookkeeping; for training speed comparisons (e.g. Figure 2) the paper only compares LSTM and BN-LSTM by iteration count; given the additional complexity of the BN-LSTM I would have also liked to see a wall-clock comparison.

As RNNs are used across many tasks, this work is of interest to many.  However, the results gains are generally minor and require several tricks to work in practice. Also, this work doesn’t address a question about batch normalization that it seems natural that it helps with faster training, but why would it also improve generalization? 

Clarity
The paper is overall very clear and well-motivated. The model is well described and easy to understand, and the plots illustrate the points clearly.

Summary
Interesting though relatively incremental adaptation, but shows batch normalization to work for RNNs where previous works have not succeeded. Comprehensive set of experiments though it is questionable if the empirical gains are significant enough to justify the increased model complexity as well as computational overhead.

Pros
- Shows batch normalization to work for RNNs where previous works have not succeeded
- Good empirical analysis of hyper-parameter choices and of the activations
- Experiments on multiple tasks
- Clarity

Cons
- Relatively incremental
- Several ‘hacks’ for the method (per-time step statistics, adding noise for exploding variance, sequence-wise normalization)
- No mention of computational overhead
- Only character or pixel-level tasks, what about word-level?

[Official Review · AnonReviewer3 · rating 8 · confidence 4 · 17 Dec 2016]

This paper extends batch normalization successfully to RNNs where batch normalization has previously failed or done poorly. The experiments and datasets tackled show definitively the improvement that batch norm LSTMs provide over standard LSTMs. They also cover a variety of examples, including character level (PTB and Text8), word level (CNN question-answering task), and pixel level (MNIST and pMNIST). The supplied training curves also quite clearly show the potential improvements in training time which is an important metric for consideration.

The experiment on pMNIST also solidly shows the advantage of batch norm in the recurrent setting for establishing long term dependencies. I additionally also appreciated the gradient flow insight, specifically the impact of unit variance on tanh derivatives. Showing it not just for batch normalization but additionally the "toy task" (Figure 1b) was hugely useful.

Overall I find this paper a useful additional contribution to the usage of batch normalization and would be necessary information for successfully employing it in a recurrent setting.

[Official Review · AnonReviewer1 · rating 7 · confidence 4 · 20 Dec 2016 (modified: 19 Jan 2017)]
**Batch normalisation brought to LSTM**

The paper shows that BN, which does not work out of the box for RNNs, can be used with LSTM when the operator is applied to the hidden-to-hidden and the input-to-hidden contribution separately. Experiments are conducted to show that it leads to improved generalisation error and faster convergence.

The paper is well written and the idea well presented. 

i) The data sets and consequently the statistical assumptions used are limited (e.g. no continuous data, only autoregressive generative modelling).
ii) The hyper parameters are nearly constant over the experiments. It is ruled out that they have not been picked in favor of one of the methods. E.g. just judging from the text, a different learning rate could have lead to equally fast convergence for vanilla LSTM. 

Concluding, the experiments are flawed and do not sufficiently support the claim. An exhaustive search of the hyper parameter space could rule that out.

[Final Decision · Program Chairs · 06 Feb 2017]
**ICLR committee final decision**

The reviewers believe this paper is of significant interest to the ICLR community, as it demonstrates how to get the popular batch normalization method to work in the recurrent setting. The fact that it has already been cited a variety of times also speaks to its interest within the community. The extensive experiments are convincing that the method works. One common criticism is that the authors don't address enough the added computational cost of the method in the text or empirically. Plots showing loss as a function of wall-clock time instead of training iteration would be more informative to readers deciding whether to use batch norm. 
 
 Pros: 
 - Gets batch normalization to work on recurrent networks (which had been elusive to many)
 - The experiments are thorough and demonstrate the method reduces training time (as a function of training iterations)
 - The paper is well written and accessible
 
 Cons
 - The contribution is relatively incremental (several tweaks to an existing method)
 - The major disadvantage to the approach is the added computational cost, but this is conspicuously not addressed.